# Evaluation and Treatment of Pain in Fetuses, Neonates and Children

**DOI:** 10.3390/children9111688

**Published:** 2022-11-03

**Authors:** Santiago Mencía, Clara Alonso, Carmen Pallás-Alonso, Jesús López-Herce

**Affiliations:** 1Pediatric Intensive Care Service, Gregorio Marañón General University Hospital, Health Research Institute of Gregorio Marañón Madrid, 28029 Madrid, Spain; 2Departamento de Salud Pública y Maternoinfantil, Facultad de Medicina, Universidad Complutense de Madrid, 28040 Madrid, Spain; 3Carlos III Institute, 28029 Madrid, Spain; 4Department of Neonatology, 12 de Octubre University Hospital, 28041 Madrid, Spain

**Keywords:** analgesia, pain, children, analgesics, behavioral pain assessment, pain scales

## Abstract

The perception of pain is individual and differs between children and adults. The structures required to feel pain are developed at 24 weeks of gestation. However, pain assessment is complicated, especially in neonates, infants and preschool-age children. Clinical scales adapted to age are the most used methods for assessing and monitoring the degree of pain in children. They evaluate several behavioral and/or physiological parameters related to pain. Some monitors detect the physiological changes that occur in association with painful stimuli, but they do not yet have a clear clinical use. Multimodal analgesia is recommended for pain treatment with non-pharmacological and pharmacological interventions. It is necessary to establish pharmacotherapeutic protocols for analgesia adjusted to the acute or chronic, type and intensity of pain, as well as age. The most used analgesics in children are paracetamol, ibuprofen, dipyrone, opioids (morphine and fentanyl) and local anesthetics. Patient-controlled analgesia is an adequate alternative for adolescent and older children in specific situations, such as after surgery. In patients with severe or persistent pain, it is very important to consult with specific pain services.

## 1. Introduction

Pain can be defined as “an unpleasant sensory and emotional experience associated with actual or potential tissue damage or described in terms of such damage” [1]. Pain consists of two features: nociception and emotional reaction.

Perception of pain is individual and differs between children and adults. The sensation of pain is not only influenced by neurophysiological mechanisms but also by both psychological aspects and the environment [2]. These aspects affect and modulate the nociceptive sensation so that the same pathological situation may cause very different painful perceptions depending on the individual. In general, children pay maximum attention to pain, which could lead increased anxiety and fear of the painful sensation, magnifying the sensory experience [3,4].

Therefore, different pain management strategies are required in children than in adults, highlighting the importance of prior preparation and non-pharmacological interventions before performing any painful procedure in a child [5,6] (Table 1).

## 2. Fetal Pain

### 2.1. Fetal Pain

Pain is a conscious and subjective experience and not just a response to noxious stimuli. For human beings to be able to experience pain, a series of physiologically mature neurological structures is needed. On the other hand, for the experience of pain to occur, other cognitive processes related to the state of consciousness and memory must be developed, which, in turn, allow an event to be discriminated as painful [7,8,9,10,11].

The development of neural pathways involved in pain pathophysiology begins early in fetal life, around the seventh week of gestation, followed by the development of the thalamus and neural connections in the cerebral cortex [7,8,9]. Therefore, some authors suggest that fetal pain does not occur before 24 weeks of gestation because the structures of the central nervous system (CNS) required for pain perception. such as the cortex, spinal cord and thalamus are not fully developed [10]. However, other authors argue that pain perception can occur, mediated by developmental structures, such as the subplate, between 12 and 20 weeks of gestation [11,12,13,14]. Additionally, behavioral changes associated with pain, such as simple motor responses, including crying and facial expressions, are described in fetuses and very premature neonates [15,16].

### 2.2. Fetal Pain Evaluation

Several physiological reactions in fetuses, such as crying, avoidance or changes in the levels of stress hormones, can be interpreted as signals of pain.

Magnetic resonance imaging (MRI) and fetal magnetoencephalographic imaging showed evoked responses to vibroacoustic and visual stimuli in the third trimester [17,18,19].

Ultrasound investigation has enabled the acquisition of behavioral characteristics in fetuses, such as crying [20,21], as well as fetal facial expressions of acute pain in surgery [22], as well as fetal movement in response to contact with an amniocentesis needle [23].

Fetuses exposed to a prolonged painful invasive procedures have increased concentrations of cortisol and beta endorphins in plasma [24,25].

### 2.3. Treatment of Fetal Pain in Fetal Surgery

Treatment of fetal pain is especially significant in fetal surgery. There are three main administration routes of fetal anesthesia and analgesia: uteroplacental transfer; intravenous, usually by the umbilical line; or intramuscularly [26]. Volatile anesthetics and opioids limit the fetal stress response, although they can produce cardiovascular fetal depression, although likely without side effects for short procedures [27].

Most authors use deep general maternal anesthesia in ex-utero intrapartum therapy surgery to anesthetic the mother and the fetus and in fetoscopies the preferences are administrate the anesthesia and analgesia directly to the fetus, usually applying an intramuscular route or umbilical line [28,29,30,31,32,33,34,35,36].

## 3. Neonatal Pain

After painful stimulus, newborns have demonstrated in MRI scans that brain regions encoding sensory and affective components of pain responses are similar to those in adults [37].

Excessive or maintained pain exposure can be detrimental, causing adverse physiological effects and even long-term consequences [38,39,40]. Preterm infants, after experiencing pain, were reported to suffer hyperalgesia and allodynia, producing prolonged stress [41].

Infants born very preterm (<32 weeks), have received many pain-related insults in a vulnerable cerebral period, require special attention [38]. Repeated pain-related stress in very premature newborns is associated with alteration of brain development during the neonatal period, [42] as well as later functional cortical activity impairment with thinning of the brain cortex, white matter microstructure alterations and cognitive outcome at school age [43,44].

## 4. Pain in Children

Children usually feel pain differently than adults. The American Academy of Pediatrics (AAP) and the American Pain Society provided a general definition of pediatric pain: “the concept of pain and suffering goes far beyond a simple sensory experience. There are emotional, cognitive and behavioral components, along with developmental, environmental and sociocultural aspects” [45]. This definition underlines the importance of the subjectivity of pain [46]. Fear and anxiety produce suffering and increase the perception of pain in children, especially the fear of separation from their parents. An important goal of pain management is to eliminate the suffering associated with pain.

## 5. Pain Assessment

### 5.1. Concepts

The first step in the treatment of pain is its detection; several circumstances must first be taken into account:-The characteristics of the child: age, sex, sociocultural level and mood.-The characteristics of the pain: form of onset, intensity, evolution, duration, etiology and consequences that may be triggered [47].

The assessment of pain in children is complex, specifically in neonates, infants and preschool-aged children, because the expression of pain is undifferentiated, so it is often not possible to distinguish between pain, irritability and anxiety. Because they are not verbal, pain assessment tools rely on surrogated measures of physiologic, behavioral and biobehavioral responses to pain.

The most frequent physiologic indicators of pain are changes in heart and respiratory rate, blood pressure and oxygen saturation. The use of vital signs alone is not adequate because neonates and infants cannot maintain an autonomic response to pain and other factors, such a mechanical ventilation and drugs [48]. These indicators are also affected by other physiological stimuli, such a hypovolemia or fever.

The most used indicators are crying, facial activity, body movements, resting positions, agitation, consolability and sleeplessness. The assessment of these behavioral indicators depends on gestational age, mechanical ventilation and pharmacological treatment, and neurologic impairment and neuromuscular blockade may also decrease or alter responses to pain in critically ill children.

### 5.2. Neonatal Pain Assessment Tools

Several neonatal pain assessment tools are available [49,50,51]. Only three pain scales, PIPP-R, N-PASS and BPSN, are adapted to premature infants. The most commonly used pain scales are summarized in Table 2 [48,52]. Pain assessment tools should be selected with consideration of the population (full-term vs. preterm), context and type of pain (procedural vs. postoperative) [51,52].

Most tools adequately assess acute pain but not persistent or prolonged pain. Only two scales are adequate for prolonged pain (N-PASS and EDIN) [53]. The COMFORTneo scale was reported to be useful for evaluation of prolonged pain in [48].

Most parameters evaluated by these tools are subjective and require observation and recording in real time.

Other tools, such as neuroimaging (functional magnetic resonance imaging and near-infrared spectroscopy) and neurophysiologic techniques (amplitude-integrated electroencephalography, changes in skin conductance and heart rate variability) during acute or prolonged pain, were studied in [44,51]. Hormonal markers of stress, such as cortisol and parameters of oxidative stress, increase with pain stimuli, but they are not used in clinical settings at this time.

### 5.3. Pain Assessment Tools for Children

#### 5.3.1. Clinical Scales

Clinical scales are the most commonly used instruments for pain assessment and monitoring [54]. The most frequently used are numerical, visual analogue or graphic scales adapted to the patient’s age. Through these scales, the patient can indicate the intensity of pain or the observer estimates the pain intensity based on the child’s behavior.

In the preverbal stage (1 month to 3 years), the scales mainly use facial expression and motor and physiological responses, such as crying. In the verbal stage (3 to 8 years) self-report can be tested using photographs and drawings of faces. From the age of 8 years onwards, verbal scale, numerical scale and graphic scales, as well as the visual analogue scale, can be used [55,56,57].

The most commonly used scales for the assessment of pain in child who are able to communicate are:The visual analogue scale (VAS), which is represented on a 10 cm line between 0 (no pain) and 10 (worst pain imaginable). VAS < 4 indicates mild or mild–moderate pain, 4–6 indicates moderate–severe pain and >6 indicates severe pain.The verbal numeric scale (VNS): the child expresses their perception of pain from 0 (no pain) to 10 (worst pain imaginable).Graphic scales, which may consist of drawings of happy faces that change to sad according to the degree of pain, columns or thermometers that are more or less filled in, color ranges, etc.

In most children admitted to pediatric intensive care units (PICUs), it is not possible to use such scales, as many of the patients are sedated and unable to communicate. The same applies to children under 3 years of age in the preverbal stage. In such cases, the identification of pain requires tools based on changes in physiological parameters, facial expression or motor response to estimate the degree of pain [54,58].

Two scales have been validated for use in critically ill children:The FLACC scale (face, leg, activity, cry, consolability) (Table 3), which considers facial expression, leg attitude, spontaneous activity, the presence and characteristics of crying and the ability to comfort or consolability, with scores ranging between 0 and 2 points for each item. A value of 0 indicates no pain, and scores of 9–10 indicate unbearable pain.

2.MAPS scale (multidimensional assessment pain scale) (Table 4). This scale is based on the observation of body movements and facial expression. It is a multidimensional scale that also includes physiological parameters, such as breathing, changes in blood pressure (BP) and heart rate (HR). Similar to the FLACC scale, it classifies pain on a scale from 0 (no pain) to 9–10 (unbearable pain) [59].

#### 5.3.2. Objective Pain Monitoring

Some monitors detect physiological changes that occur in association with painful stimuli and stress based on electromyograms, plethysmography, electrocardiograms, skin conductance or measurement of the diameter of the pupil. Such monitors can quantify the intensity of pain and the response to treatment. These parameters are objective, non-invasive, relatively easy to use and can be used at the patient’s bedside. However, they are unspecific (other non-pain stimulus, such as agitation, fear and stress, produce the same response) and are expensive. Therefore, they have not yet been applied in clinical settings [60,61]. Such monitors include:

Conductance skin impedance monitor (Medstorm Innovations, Oslo, Norway), which assesses the stress response through changes in skin conductance produced by sympathetic stimulation. It is non-invasive with rapid response but does not differentiate between agitation and pain and it is not able to measure baseline stress [62].

The analgesia nociception index (ANI), which is based on the RR variability of ECG intervals. This is a continuous line measurement of parasympathetic tone, which is part of the autonomic nervous system that assesses nociception [63,64].

The pain pupillary index (PPI) measurement, which provides real-time information from a video camera and infrared pupillary diameter [65].

## 6. Treatment

In the 2010 Montreal Declaration, access to pain treatment was declared a fundamental human right, constituting a violation of human rights not to treat pain [66,67,68,69,70,71].

It is important to develop treatment protocols to prevent procedural pain by combining pharmacological and non-pharmacological strategies. The intensity of analgesia should be adjusted to the intensity of the potential pain and ensure adequate supervision and monitoring [72,73].

### 6.1. Non-Pharmacological Analgesia

The aim of non-pharmacological analgesia is to increase patient comfort and reduce stress related to diagnostic or therapeutic procedures [74,75]. It should not be used as a substitute for pharmacological treatment but should be combined with pharmacological treatment [76,77,78].

Mechanisms of action are not yet well explained, but such interventions likely reduce the sensitivity of the nociceptive system. Some such measures release endogenous endorphins, activating opioid-enhancing neuropeptides inducing distraction from pain.

Such measures can be classified based on the mechanism of action [79].

Environmental strategies: low noise and lighting, soothing smells or clustering procedures to avoid over handling by modifying the environment to reduce sensibility to pain.

Cognitive strategies, including distraction methods, particularly used in older children.

Behavioral strategies, including direct (rocking) or indirect (non-nutritive sucking) manipulation of the infant’s body. These techniques produce relaxing tactile stimuli before, during and/or after the painful procedure to reduce pain [80,81].

The main non-pharmacological strategies used in neonates and children are:

Posture and mobilization: Mobilizations are applied using pillows or devices to help increase well-being [79]. This strategy is effective in term and preterm babies.

Breastfeeding or expressed breast milk: breastfeeding [82] is one of the most effective non-pharmacological strategies. Supplemental breast milk is less effective than breastfeeding and sucrose [83,84].

Oral sucrose and glucose [85]: It remains unclear whether sucrose suppresses the responses to pain. It should be administered 2 min before the procedure, and the effect lasts approximately 4 min [86].

Non-nutritive sucking [79], which can be achieved with a pacifier, breastfeeding after extraction or with the finger. It is effective in term and preterm babies.

Skin-to-skin contact (kangaroo care) [87], which is effective for pain relief in neonates.

Sensorial saturation [88], which results from multimodal sensory inputs (e.g., touch, massage, taste, voice and smell) during a painful procedure.

Distraction: attracting attention, keeping thoughts occupied and away from pain (music, images, games, etc.) [89].

Cognitive reformulation: in older children, recognize thoughts that increase pain and replace them with positive thoughts.

Music therapy [90]: it can improve heart rate, feeding and sucking in preterm infants.

Mother’s voice: some studies have demonstrated analgesic qualities in neonates that resulting from exposure to the mother’s voice [91].

Other measures studied for the relief pain include massage therapy [92], medical acupuncture [93,94], osteopathic manipulation [95] and radiant warming [96] or localized warming for painful procedures.

The main advantages of non-pharmacological treatments include ease of learning and performance, as well as safety and feasibility. However, no studies have been conducted to date analyzing the long-term effects of these techniques [81].

### 6.2. Pharmacological Analgesia

#### 6.2.1. Concepts

The responsible physician must establish the treatment, depending on the age, type and intensity of the pain (Figure 1 and Figure 2), as well as the underlying disease, and assess the following aspects:Patient’s age and associated pathology (cardiorespiratory, renal and hepatic function);Check for interactions of analgesics with other medications the patient is receiving; andPrevent and treat the most common side effects of analgesics [97].

An analgesic regimen and analgesic rescue should be prescribed according to the intensity of pain [98] (Figure 2). Pharmacotherapeutic protocols for pediatric pain need to be established according to intensity, age and the form of drug administration [99,100,101,102].

#### 6.2.2. Pharmacological Analgesia in Neonates

The most used pharmacological analgesia in neonates are paracetamol, opioids and local anesthetics [44,103]. The doses and effects of most frequent drugs used in neonates are summarized in Table 5 [104,105,106,107,108,109,110,111,112].

Pharmacological pain management also involves risks of adverse effects, such as respiratory depression or neurotoxicity. This association is particularly evident for prolonged or repeated exposure to analgosedative drugs [38,104].

Opioids should not be administered routinely in neonates with mechanical ventilation, owing to the possible neurotoxicity of opioids in the developing brain [38,104,105,106]. Opioid administration should be based on clinical evaluation and scales [107]. The use of combined analgesic regimens permits a reduction in the dose of opioids by maintaining adequate analgesia with reduced frequency of side effects [107,108].

Current guidelines promote a pharmacological therapy administered in a stepwise approach, together with regular assessment of pain and sedation scores [53,109,110,111,112].

#### 6.2.3. Pharmacological Analgesia in Children

A variety of analgesic drugs is available [113,114,115,116,117,118,119]. Table 6 summarizes the drugs, doses and effects of the most commonly used analgesics in pediatrics [98,99,100,101,108,113,114,115,116,117,118,119,120,121,122]. Pharmacological analgesia in children depends on the procedure, kind and intensity of pain and the age of the patient (Table 7) [120,121,122].

### 6.3. Local Analgesia

Multiple formulations of topical anesthetics are available for use in children. The most used are included in Table 8 [123,124,125,126,127].

### 6.4. Other Methods

Patients with severe pain that is not well controlled despite powerful opioids at full dosages can become candidates for special anesthesia and regional analgesia techniques [94]. Patients with high tolerance to opioids, severe adverse effects, chest pain with respiratory compromise or chronic pain may also be candidates for these techniques.

#### 6.4.1. PCA Pumps

From the age of 6–7 years, well-educated children with adequate cognitive abilities are able to self-administer analgesics by means of a PCA-programmed device [128]. The most commonly used modality in pediatrics is the mixed mode, whereby the pump administers a minimal basal continuous infusion, and if this is not sufficient, the patient can self-administer boluses until pain is eliminated or reduced. The system has important psychological advantages, especially for adolescents, by improving the perception of control over their own pain and reducing unnecessary analgesic consumption and side effects of opioid drugs. The most commonly used opioids in pediatrics are morphine, fentanyl, hydromorphone, oxycodone and tramadol. In pediatric patients under 6 years of age or without adequate cognitive capacity, the “PCA by proxy” modality can be used, whereby the nurse administers the analgesic rescue boluses (“nurse PCA”) or even selected parents (“parent PCA”).

#### 6.4.2. Regional and Interventional Analgesia Techniques

Regional techniques have been applied in children with satisfactory results [129,130,131]. The most common technique is epidural analgesia, which can be given at any level of the neuroaxis and which, by insertion of a catheter, allows analgesia to be maintained for several days or even weeks if necessary. In such cases, the catheter must be tunneled. Other catheter-based analgesia techniques, such as paravertebral, fascial and incisional analgesia use, have increased in popularity in pediatric patients [128] (Table 9). In all such techniques, local anesthetics are administered alone or in combination with adjuvants, such as opioids, clonidine, dexmedetomidine or corticosteroids, which enhance their duration and effect. Regional and invasive techniques provide pain elimination without causing CNS side effects, such as those produced by opioids.

## 7. Pain Assessment and Treatment Algorithm

The objective of the evaluation of pain and stress is to detect pain and determine the appropriate analgesic dose for each patient without side effects of treatment.

The first step to individualize the treatment of each patient is to implement a validated pain assessment tool, usually clinical scales (the fifth vital sign).

The second step is to evaluate other indicators of the patient, environment and therapies. To establish the treatment of pain, the assessment of pain, the environment, the individual characteristics of the patient, the underlying disease and other treatments must be taken into account.

It is not helpful to implement pain assessment as a stand-alone procedure. Therefore, pain assessment must be followed by a stepped treatment protocol adapted to the patient, as well as a structured follow-up and with multidisciplinary responsibility.

It is recommended to perform multimodal analgesia with non-pharmacological and/or pharmacological interventions. Subsequently, the patient must be re-evaluated to check the response and adjust the medications at the lowest dose necessary at each moment.

Figure 1 and Figure 2 summarize the algorithm for the assessment and treatment of pain in neonates and children.

## 8. Conclusions

Pain assessment is complicated, especially in neonates, infants and preschool-aged children. We recommend using clinical scales adapted to the age of the patient to assess and monitor the degree of pain in children.

We recommend multimodal analgesia with non-pharmacological and pharmacological interventions for pain treatment and the establishment of pharmacotherapeutic protocols for analgesia adjusted to the acute or chronic, type and intensity of pain, as well as the age of the patient. Patient-controlled analgesia is an adequate alternative for adolescents and older children in specific situations, such as after surgery. Patients with severe or persistent pain should be treated in consultation with specific pain services.

The follow-up of structured protocols will enable early diagnosis of pain and treatment adapted to the intensity and characteristics of the pain and of the child, which can reduce suffering and contribute improved prognosis.

## Figures and Tables

**Figure 1 children-09-01688-f001:**
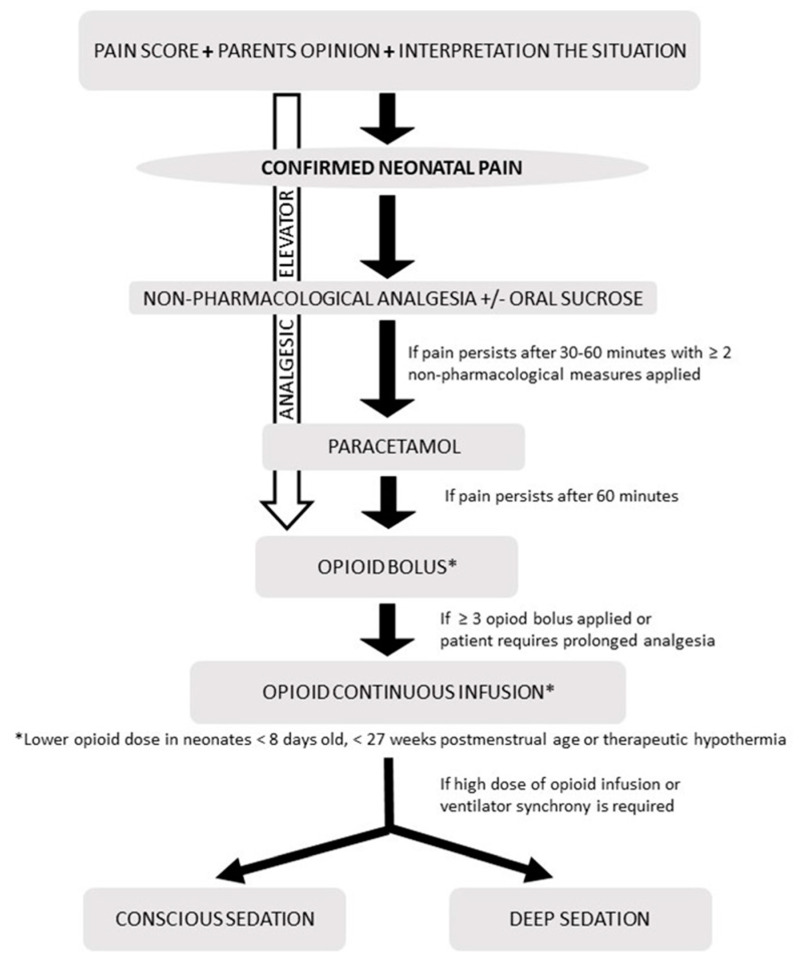
Algorithm for pain assessment and management in neonates.

**Figure 2 children-09-01688-f002:**
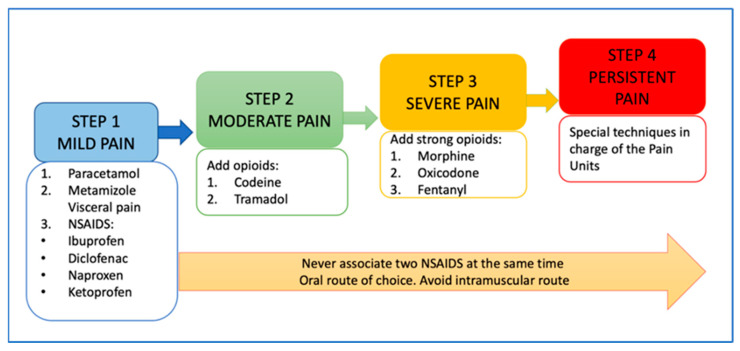
The analgesic ladder proposed by the World Health Organization (WHO) for children [96].

**Table 1 children-09-01688-t001:** Pain differences according to age.

	Neonates and Infants	Children	Adults
Development of pain sensitivity	Yes, undefined	Yes, defined	Yes, defined
Location of pain	No	No; Yes in older children	Yes
Verbal expression	Cry	Verbal; Could be non-specific	Verbal and specific
Pain response	Important and generalized (tachycardia, tachypnea, hypertension–hypotension, severe stress and agitation)	Important in toddlers;Moderate in older children	Mild
Difference between anxiety and pain	No	Only older children	Yes
Clinical evaluation	Physiological scales	Physiological and verbal scales, non-numerical	Numerical scales
Analgesic drugs	Very few studies; Empirical off-label treatment for most drugs; Undefined doses	Few studies; Empirical off-label treatment for most drugs	Many studies; Well-defined doses
Side effects of analgesic drugs	Severe systemic side effects (bradycardia, hypotension and respiratory arrest) Psychomotor development effect?	Moderate systemic side effects Psychomotor development effect?	Mild systemic side effects

**Table 2 children-09-01688-t002:** Neonatal pain scales.

Pain Scale	Gestational Age	Parameters	Type of Pain	Scale Metric
PIPP-R (premature infant pain profile-revised)	26 weeks to term	Heart rate and oxygen saturation. Alertness, brow bulge, eye squeeze and nasolabial furrow	Procedural and postoperative	0–21
CRIES (cries, requires oxygen, increased vital signs, expression, sleeplessness)	32–56 weeks	Blood pressure, heart rate, oxygen saturation. Cry, expression and sleeplessness	Postoperative	0–10
NIPS (neonatal infant pain scale)	28–38 weeks	Breathing pattern. Facial expression, cry, arms, legs and alertness	Procedural	0–7
COMFORT neo	24–42 weeks	Respiratory response, blood pressure and heart rate. Alertness, agitation, physical movements, muscle tone and facial tension	Prolonged	8–40
NFCS (neonatal facial coding system)	25 weeks to term	Brow bulge, eye squeeze, nasolabial furrow, open lips, stretched mouth, lip purse, taut tongue and chin quiver	Procedural	0–10
N-PASS (neonatal pain, agitation and sedation scale)	0–100 days	Heart rate, respiratory rate, blood pressure and oxygen saturation. Crying or irritability, behavior state, facial expression, extremities or tone	Acute and prolonged pain. Also assesses sedation	Pain 0–10 Sedation −10–0
EDIN (échelle de la douleur inconfort noveau-né)	25–36 weeks	Facial activity, body movements, quality of sleep, quality of contact with nurses and consolability	Prolonged	0–15
BPSN (Bernese pain scale for neonates)	27–41 weeks	Respiratory pattern, heart rate and oxygen saturation. Alertness, duration of cry, time to calm, skin color, brow bulge with eye squeeze and posture	Procedural	0–27

**Table 3 children-09-01688-t003:** FLACC scale (face, leg, activity, cry, consolability).

	0	1	2
**Face**	No particular expression or smile	Occasional grimace or frown, withdrawn, disinterested.	Frequent to constant quivering chin, clenched jaw
Legs	Normal position or relaxed	Uneasy, restless, tense	Kicking or legs drawn up
Activity	Crying quietly, normal position, moves easily.	Squirming, shifting back and forth, tense.	Arched, rigid or jerking.
Cry	No cry (awake or asleep)	Moans or whimpers; occasional complaint.	Crying steadily, screams or sobs, frequent complaints.
Consolability	Content, relaxed.	Reassured by occasional touching, hugging or being talk to; distractible.	Difficult to console or comfort.

No pain, 0; mild pain, 1–2; moderate pain, 3–5; severe pain, 6–8; extreme–maximum pain, 9–10.

**Table 4 children-09-01688-t004:** MAPS scale (multidimensional assessment pain scale).

	0	1	2
**Vital signs: HR and/or BP**	Within baseline	More than 10 bpm increase and/or more than 10 mmHg increase	More than 10 bpm decrease and/or more than 10 mmHg decrease
Breathing pattern	No change	Development or increase in respiratory distress	Severe respiratory distress
Facial expression	Relaxed	Grimace	Grimace associated with silent or weak cry
Body movements	No movements or purposeful movements	Restless	Rigid and/or limited body movements
State of arousal	Calm or asleep	Hyperreactive	Shut down

No pain, 0; mild pain, 1–2; moderate pain, 3–5; severe pain, 6–8; extreme–maximum pain, 9–10.

**Table 5 children-09-01688-t005:** Pharmacological agents for neonatal pain management.

Drug	Dose	Pharmacology	Indications	Adverse Effects
Morphine	Boluses: 30–50 mcg/kg, maximum 100 mcg/kg over 15–30 min Infusion: 30–100 mcg/kg bolus followed by 10 mcg/kg/h A lower initial dose is recommended. Therapeutic hypothermia: Bolus of 50 mcg/kg, followed by 5 mcg/kg/h infusion	Action of 5 min. Peak effectiveness: 15 min. Half-life: 6–12 h	Analgesia and sedation in mechanical ventilation; postoperative pain control; sedation during therapeutic hypothermia. Not be used in preterm infants less than 27 weeks and neonates with hemodynamic instability	Hypotension, respiratory arrest, urinary retention, tolerance, withdrawal, risk of periventricular leukomalacia and/or death; poor long-term neurodevelopmental outcomes
Fentanyl	Bolus of 0.5–2.0 μg/kg Infusion: 0.5–2.0 μg/kg/h Preterm neonates born before 32 w g reduced by 50% during days 0–4, Therapeutic hypothermia 1–2 mcg/kg over one hour and then infusion to 0.5–1 mcg/kg/h	Action of 1–2 min. Duration of action: 60 min. Half-life: 3.1–9.5 h	Short procedures. Analgesia and sedation in mechanical ventilation; sedation during therapeutic hypothermia	Respiratory arrest, chest wall rigidity, laryngospasm, tolerance withdrawal, delayed meconium passage. Cerebellar hypoplasia; decrease in eye and hand coordination skills at two years
Remifentanil	Endotracheal intubation:1–2 μg/kg. Percutaneous central venous catheter: 0.25 μg/kg/min Sedation and analgesia in mechanical ventilation: 0.15 μg/kg/min	Action: 1 min Half-life: 3.5–5 min.	Short procedures: endotracheal intubation, laser surgery for retinopathy of prematurity	Bradycardia, hypotension, chest wall rigidity, tolerance and withdrawal; no studies of long-term outcome in neonates
Paracetamol	Oral dose: 25 mg/kg/day in at 30 w, 45 mg/kg/day at 34 w, 60 mg/kg/day in term n Intravenous: 20 mg/kg loading dose, followed by a 10 mg/kg maintenance dose every six hours. 28 to 31 wg, to 12 h	IV action: 5 min. Peak effectiveness: 15 min Oral peak effectiveness: 1 h Enteral and rectal: variable absorption	Mild to moderate pain;reduction in morphine requirements after major surgery	Hardly causes hepatic or renal toxicity in newborns; minor hemodynamic effects have been found following IV; Following neonatal exposure to paracetamol remains limited; possible link with risks for atopy, fertility and neurobehavioral problems
Dexmedetomidine	Bolus 0.05–0.2 mg/kg over 15 min followed by maintenance 0.47 ± 0.21 mg/kg/h Therapeutic hypothermia 0.3 mcg/kg/h (range 0.2–0.5 mcg/kg/h)	Half-life: 7.6 h preterm and 3.2 h in term infants	Alternative agent for sedation in mechanical ventilation. Sedation of term neonates during therapeutic hypothermia; no studies in neonates	Hypotension and bradycardia generally self-limiting; reduced dosed withdrawal (weaned by decreasing the infusion by 0.1 mcg/kg/h every 12 to 24 h)
Ketamine	Bolus 1–2 mg/kg	Action:1–2 min; short duration:15–30 min.	Short painful procedures; hemodynamically unstable patients; no studies in neonates	Possible Neurotoxicity; no studies of long-term outcome in neonates

**Table 6 children-09-01688-t006:** Most commonly used analgesic drugs in pediatrics.

Drug	Dose	Indications	Comments and Main Side Effects
Acetaminophen	po/pr: 5–10 mg/kg/h IV: 10–15 mg/kg/6 h (<10 kg: 7.5 mg/kg/6 h) <1 year: 7.5 mg/kg	Moderate pain Hyperthermia	- Hepatotoxicity
Ibuprofen	po: 5–10 mg/kg/6 h IV: 5–10 mg/kg/6 h	Moderate pain Hyperthermia	- Of choice in children - Gastrointestinal bleeding
Metamizole	po: 10–15 mg/kg/6–8 h IV: 10–20 mg/kg/6–8 h	Moderate–severe pain Hyperthermia	- Synergistic effect with opioids - Bone marrow aplasia
Dexketoprofen	po: 0.5–1 mg/kg/8 h IV/im: 10–40 mg/kg/8 h	Moderate–severe pain Anti-inflammatory	- Not recommended for children under 12 years of age - Gastrointestinal bleeding
Ketorolac	po: 0.5 mg/kg/6–8 h IV, im: 0.2–1 mg/kg/6–8 h	Moderate–severe pain Anti-inflammatory	- Gastrointestinal bleeding - Nephrotoxicity
Naproxen	po/pr/im: 5 mg/kg/8–12 h	Mild–moderate pain Anti-inflammatory	- Not recommended for children under 12 years of age - No IV - Gastrointestinal bleeding
Diclofenac	po: 0.5–1.5 mg/kg/8 h	Mild–moderate pain Spasmolytic	- Spasmolytic effect - Gastrointestinal bleeding
Tramadol	po/pr/sc: 1–2 mg/kg/4–6 h iv: 1–2 mg/kg/4–6 h	Acute pain	- Good hemodynamic tolerability - Less respiratory effect
Meperidine	IV/im/sc: 0.5–2 mg/kg/4–6 h	Acute pain Spasmolytic	- Constipation - Urinary retention
Morphine	po: 0.2–0.5 mg/kg/6–8 h IV/im/sc: 0.1–0.2 mg/kg/4–6 h CII: 10–50 mcg/kg/h	Analgosedation in conventional mechanical ventilation Acute or chronic pain Pulmonary edema	- Dose adjustment in renal and hepatic failure - Release of histamine, nausea, vomiting and respiratory arrest
Fentanyl	IV/sc/sl/in: 1–3 mcg/kg CII: 1–10 mcg/kg/h	Procedural pain Analgosedation in conventional mechanical ventilation Acute or chronic pain Pulmonary edema	- Long elimination - Improved cardiocirculatory stability - Chest stiffness and respiratory arrest
Nitrous oxide	20–70% with oxygen	Procedural pain Endoscopy and venipunctures	Nausea, vomiting Myocardial dysfunctionNeurodevelopment effects?

po: per os; pr: per rectal; in: intranasal; sl: sublingual; sc: subcutaneous; IV: intravenous infusion; CII: continuous intravenous infusion.

**Table 7 children-09-01688-t007:** Pain treatment according to intensity and patient age.

Age	Mild Pain	Moderate Pain	Severe Pain
**<1 year old**	**Adequate oral tolerance** - Ibuprofen 2%: 4–10 mg/kg/oral 8 h - Rescue (a) Acetaminophen 100 mg/mL: 10 mg/kg/6–8 h/oral (b) Metamizole: 12.5 mg/kg/8 h (max 500 mg/dose) **No oral tolerance** - Acetaminophen IV: 7.5–10 mg/kg/6–8 h - Rescue: Metamizole: 20 mg/kg/IV/8 h	**Adequate oral tolerance** - Ibuprofen 2%: 4–10 mg/kg/oral 8 h + acetaminophen 100 mg/mL: 10 mg/kg/6–8 h/oral. - Rescue: Metamizole: 20 mg/kg/8 h (max 500 mg/dose). **No oral tolerance** - Acetaminophen IV: 7.5–10 mg/kg/iv/6–8 h + metamizole: 20 mg/kg/IV/8 h - Rescue: pethidine 0.5 mg/kg/IV/6 h	**Intravenous** - Metamizole 100 mg/kg/IV/24 h CII + morphine: 0.1–0.25 mg/kg/IV/6 h - Rescue: (a) Acetaminophen 7.5–10 mg/kg/IV/6 h (b) PCA IV: fentanyl + metamizole Continuous respiratory rate and oximetry monitoring
**>1 year old**	**Adequate oral tolerance** - Ibuprofen 2%: 4–10 mg/kg/oral 8 h - Rescue: (a) Acetaminophen 100 mg/mL: 10–15 mg/kg/6–8 h/oral (b) Metamizol: 12.5 mg/kg/8 h (max 500 mg/dose) **No oral tolerance** - Acetaminophen IV: 10–15 mg/kg/6–8 h - Rescue: Metamizol IV 20 mg/kg/8 h	**Adequate oral tolerance** - Ibuprofen 2%: 4–10 mg/kg/oral 8 h + acetaminophen (100 mg/mL): 15 mg/kg/6–8 h/oral. - Rescue (a) Metamizole: 20 mg/kg/8 h (max 500 mg/dose). (b) Tramadol 1 mg/kg/6 h (max 8 mg/kg/24 h) **No oral tolerance** - Acetaminophen IV 10–15 mg/kg/6 h + metamizol IV 20–40 mg/kg/8 h - Rescue: (a) Morphine 0.1–0.25 mg/kg/IV/6 h (b) Pethidine 0.5 mg/kg/IV/6 h	**Intravenous** - Metamizole 120 mg/kg/IV/CII in 24 h + morphine: 0.1–0.25 mg/kg/6h - Rescue (a) Acetaminophen 15 mg/kg/IV/6 h (b) PCA IV fentanyl + metamizole Continuous respiratory rate and pulse oximetric monitoring.

**Table 8 children-09-01688-t008:** Local analgesia in newborns and children.

Drug	Dose	Indications	Adverse Effects
EMLA (lidocaine 2.5%-prilocaine 2.5%).	0.5 g to 1.0 g applied to the procedural site; anesthesia within 60–90 min	Lumbar puncture, venipuncture, circumcision	Methemoglobinemia
Tetracaine gel.	Applied to the procedural site	Intramuscular injection and heel sticks	Transient local erythema
Liposomal lidocaine 4% cream (LMX4)	Applied to the procedural site; anesthesia within 30 min	Venipuncture, skin biopsies, lesion removal and electrocautery	No risk of methemoglobinemia
Lidocaine	Subcutaneous infiltration; 0.5 mL/kg of 1% or 0.25 mL/kg of 2%	Lumbar puncture, circumcision, percutaneous venous or arterial catheter placement	In neonates, avoid combination with epinephrine (risk of tissue necrosis and tachyarrhythmias)
Proparacaine anesthetic eye drops (alcaine) 0.5%	30 s before eye examination	Retinopathy of prematurity screening	Eye redness

**Table 9 children-09-01688-t009:** Interventional techniques for pediatric pain.

Technique	Pediatric Experience	Indications	Guided
Neuroaxial block	Very extensive	Oncological pain, neuropathic pain, complex regional pain, refractive phantom limb	Loss of resistance US, Radioscopy
Peripheral nerve block	Extensive	Oncological pain, neuropathic pain, complex regional pain, refractive phantom limb Chronic abdominal wall pain	US Electric stimulation
Sympathetic block	Less extensive	Complex regional pain, herpes, visceral pain	US, Radioscopy
Major occipital nerve	Less extensive	Occipital neuralgia, post-traumatic headache, migraine	US
Fascial blocks: TPA (transversal plane abdomen), rectus fascia, iliac fascia, and ilioinguinal fascia	Less extensive	Abdominal wall pain Post herniorrhaphy neuralgia Cutaneous nerve entrapment syndrome Myofascial pain	US
IV block (Bier)	Less extensive	Neuropathic pain, complex regional pain	
Chemical neurolysis	Short experience	Oncological pain spasticity	Radioscopy, US
Radiofrequency	Short experience	Refractory neuropathic pain, joint pain, oncological pain	
Transcutaneous electrical nerve stimulation (TENS); spinal cord, brain	Very extensive Less extensive; very short experience	Neuropathic pain, complex regional pain	Radioscopy, US Radioscopy + surgery
Intrathecal baclofen	Extensive	Infantile spastic cerebral palsy, dystonia	Radioscopy, US
Intra-articular	Extensive	Juvenile idiopathic rheumatoid arthritis, spondylitis	US

CAT: US: Ultrasound.

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
