# Peer review of "Evaluation and Treatment of Pain in Fetuses, Neonates and Children"

_children, 2022, doi:10.3390/children9111688_

Round 1
Reviewer 1 Report
The manuscript summarized the current state of pain evaluation methods and treatments, especially for fetuses, neonates, and children. They first defined and collected the facts of pain for different ages, introduced current assessments of pain, and gave some illustrations on possible treatments. Finally concluded the algorithm of pain assessment and treatment. This manuscript is well written, however, like a textbook rather than a journal article.
1. What readers do you targeting? Clinical doctors already have some guidelines for treatment so they don't need to read this manuscript. Researchers already know the current assessment tools for different needs and thus they don't need to read this manuscript. In this case, who would be the audience? If this manuscript was written for students, then you should make this article a textbook chapter rather than a review.
2. What is the goal of this manuscript? I did not see the core value of this manuscript. I believe the author wants to emphasize an idea but the idea is too implicit in the current manuscript. In section 7 (page 13), the author emphasized that detecting pain and finding an appropriate analgesic dose for each patient is important, while also mentioning that in the clinic it is not helpful to implement pain assessment as a stand-alone procedure. Maybe the author can elaborate more on this point. Alternatively, the author mentioned that children and adults feel pain in different manners (pages 1 and 2), probably the author can elaborate more on how the difference would therefore affect assessment and treatment and give some future suggestions (rather than just listing the current states).
3. The treatment section mentioned some non-pharmacological analgesia; however, were listed without categorization. For instance, cognitive strategies were mentioned again in cognitive reformulation. Sucking for different objects was mentioned several times. Distraction can be one type of cognitive strategy. Music therapy was confused with the mother's voice.
4. I would appreciate that if the author could give summary tables or figures to illustrate how pain was different and what are the major concerns in different age levels (pages 1-3).
5. Is Figure 1 made according to any guidelines? Why the step evaluation depends on persistence, rather than pain intensity? How was the pain assessed? What is the major concern in the age? What are the differences in procedure between Figures 1 and 2? Why? Similarly, would the procedure in Figure 2 apply to adults? Why and why not?
Minor points:
1. This manuscript was an article, which is misleading. There were no experiments, participants, or data analysis in this manuscript. This manuscript should be a "review" rather than an "article".
2. page 1 lines 39-40: "... may cause very different painful sensations." I would suggest replacing sensation with perception.
3. On page 5 the conductance, ANI, and PPI are in a strange editing format, please check.
4. The paper ended by introducing Figures 1 and 2, which is not a recommended ending. Some conclusion or summary is needed.
Author Response
Reviewer 1
Thank you very much for your revision and suggestions that improve the quality and presentation of our article. We response your questions and explain the changes made in the manuscript
- What readers do you targeting? Clinical doctors already have some guidelines for treatment so they don't need to read this manuscript. Researchers already know the current assessment tools for different needs and thus they don't need to read this manuscript. In this case, who would be the audience? If this manuscript was written for students, then you should make this article a textbook chapter rather than a review.
We think this article is pertinent because there no articles review pain from fetus to children. Moreover, many gynaecologists and paediatricians don´t use objective scales to evaluate pain and don´t have an integrate vision of pain and analgesic protocol. Several articles showed that, even units as Emergency an PICU that treat frequently children with pain don´t have specific protocols to evaluate and treat pain, and the quality evaluation of Children hospitals showed the same situation. In a multicenter multinational study Grunauer et al found a statistically significant more effective pain management in PICU high income countries income than in low-middle-income countries. There are many barriers that practitioners confront in everyday practice, including access to validated tools to assess and treat pain, deficient practitioner training, a lack of pain experts Front Pediatr 2021
On the other hand, this review article is not a chapter of a book because it not based in the general characteristics or physiopathology of the pain but practice protocol of evaluation and treat the pain of children
- What is the goal of this manuscript? I did not see the core value of this manuscript. I believe the author wants to emphasize an idea but the idea is too implicit in the current manuscript. In section 7 (page 13), the author emphasized that detecting pain and finding an appropriate analgesic dose for each patient is important, while also mentioning that in the clinic it is not helpful to implement pain assessment as a stand-alone procedure. Maybe the author can elaborate more on this point.
We probably have not expressed this idea well. What we mean is that the assessment of pain cannot be an independent isolated event but must be followed by a stepped treatment protocol adapted to the patient and with a structured follow-up. In accordance with the recommendations of the reviewer we have included this explanation in the text.
- Alternatively, the author mentioned that children and adults feel pain in different manners (pages 1 and 2), probably the author can elaborate more on how the difference would therefore affect assessment and treatment and give some future suggestions (rather than just listing the current states).
According to the review recommendations we have explained better the differences in pain between children and adults. We have added a new table (table 1) including these differences.
- The treatment section mentioned some non-pharmacological analgesia; however, were listed without categorization. For instance, cognitive strategies were mentioned again in cognitive reformulation. Sucking for different objects was mentioned several times. Distraction can be one type of cognitive strategy. Music therapy was confused with the mother's voice.
In the non-pharmacologic analgesia section we explain first that measures can be classified based on the mechanism of action (environmental, cognitive and behavioral). After this, we listed the most used methods. We have corrected the mistake with music and mother voice.
- I would appreciate that if the author could give summary tables or figures to illustrate how pain was different and what are the major concerns in different age levels (pages 1-3).
According to the recommendations of the reviewer we have addeed a table with the differences of pain and evaluation in different age levels (table 1)
- Is Figure 1 made according to any guidelines? Why the step evaluation depends on persistence, rather than pain intensity? How was the pain assessed? What is the major concern in the age? What are the differences in procedure between Figures 1 and 2? Why? Similarly, would the procedure in Figure 2 apply to adults? Why and why not?
Figure 1 showed the protocol of treatment of pain in neonates based in persistence. The figure shows that the evaluation of pain is based of pain scores and patient and health care opinions. We agree with the reviewer that this protocol is not bases in the intensity of the pain. On the other hand, the figure 2 classify the pain according to the intensity and also the persistence.
Minor points:
- This manuscript was an article, which is misleading. There were no experiments, participants, or data analysis in this manuscript. This manuscript should be a "review" rather than an "article".
The article is a review.
- page 1 lines 39-40: "... may cause very different painful sensations." I would suggest replacing sensation with perception.
We have replaced sensation by perception according to the suggestions of the reviewer
- On page 5 the conductance, ANI, and PPI are in a strange editing format, please check
We have checked the format of this paragraph
- The paper ended by introducing Figures 1 and 2, which is not a recommended ending. Some conclusion or summary is needed.
According to the recommendations of the reviewer we have added a paragraph as conclusion

Reviewer 2 Report
Very nice article- flows extremely well and tables are informative and easy to read. Algorithm is also excellent.
Reads well.
Congratulations
Author Response
Thank you very much for your opinion

Reviewer 3 Report
Dear Authors,
The perception of pain in children is different from that in adults, therefore the determination of the degree of pain and the treatment of pain require different solutions. In this review, the authors attempted to summarize the current solutions for the assessment and management of pain in children.
The topic is timely and important and may attract much attention. However, I have some suggestions to improve this paper:
1. Clinical scales - this chapter should be reworked a bit. It contains repetitive information and sentences that can be combined.
2. 5.3.2. chapter - do not include references in the subtitle. +text formatting error
3. 6.4. chapter - some references would be worth including.
4. Conclusion is missing - It would be worth describing in a few sentences what the observations are in clinical practice for the evaluation and management of pain. What could improve this?
5. References: In general, I recommend authors use more references to back their claims. Thus, I recommend the authors attempt to deepen the subject of their article, as the bibliography is too concise. Nonetheless, in my opinion, less than 150 articles for a review paper are insufficient. Currently, authors cite only 97 papers, therefore, I suggest the authors focus their efforts on researching relevant literature: I believe that adding more citations will help to provide better and more accurate background to this study.
6. The side effects of pharmacological treatments should be briefly described (it can even write in the tables).
7. Formal errors:
Several typing errors are present in the manuscript. Please, review the text carefully and correct these errors.
Some examples:
·space errors
Line 23 application.For
·typing errors
Line 62 fetuses. [16,17]
Line 71 procedures [22]
Line 78 adults [32]
Line 82 stress [34]
Line 115 pulmonary disease
Line 126 postoperative). [41].
Line 167 Consolability). (Table 2).
Line 173 Scale). (Table 3).
Line 247 disease. and
Line 263 anesthetics. [37,84].
·formatting errors
there are yellow highlights in several places in table 5.
8. English proofreading is recommended.
Questions
1. Hormonal and plasma markers - why are they not used in clinical practice?
2. In the case of minor pain, why can't non-pharmacological treatment be used independently?
3. What methods exist that are not yet used in clinical practice, but could help in the assessment and management of pain?
Author Response
Reviewer 3
Thank you very much for your revision and suggestions that improve the quality and presentation of our article. We response your questions and explain the changes made in the manuscript
- Clinical scales - this chapter should be reworked a bit. It contains repetitive information and sentences that can be combined.
We have deleted some repetitive information
- 3.2. chapter - do not include references in the subtitle. +text formatting error
We have deleted the references in the subtitle
- 4. chapter - some references would be worth including.
We have included some references of this part of the article
- Conclusion is missing - It would be worth describing in a few sentences what the observations are in clinical practice for the evaluation and management of pain. What could improve this?
According to the recommendations of the reviewer we have added a final conclusion
- References: In general, I recommend authors use more references to back their claims. Thus, I recommend the authors attempt to deepen the subject of their article, as the bibliography is too concise. Nonetheless, in my opinion, less than 150 articles for a review paper are insufficient. Currently, authors cite only 97 papers, therefore, I suggest the authors focus their efforts on researching relevant literature: I believe that adding more citations will help to provide better and more accurate background to this study.
We think that the number of references was sufficient to support each of part of the article. However, following the recommendations of the reviewer I have added more references.
- The side effects of pharmacological treatments should be briefly described (it can even write in the tables).
The main side effects of each drug were included in tables 4 an 5 . However, we added some more side effects.
- Formal errors:
Several typing errors are present in the manuscript. Please, review the text carefully and correct these errors.
I have corrected these mistakes
Some examples:
- space errors
Line 23 application.For
- typing errors
Line 62 fetuses. [16,17]
Line 71 procedures [22]
Line 78 adults [32]
Line 82 stress [34]
Line 115 pulmonary disease
Line 126 postoperative). [41].
Line 167 Consolability). (Table 2).
Line 173 Scale). (Table 3).
Line 247 disease. and
Line 263 anesthetics. [37,84].
- formatting errors
there are yellow highlights in several places in table 5.
- English proofreading is recommended.
The text has been reviewed by an English-speaking doctor
Questions
- Hormonal and plasma markers - why are they not used in clinical practice?
Hormonal and plasma markers are unspecific and lab results are late. We need rapid evaluation methods at the patient´s bedside. For this reason, they are not used in clinical practice.
- In the case of minor pain, why can't non-pharmacological treatment be used independently?
I case of minor and short-term pain non-pharmacological treatment can be used. These methods are used for some procedures like venous punction. For minor and long-term pain, the combination of some non-pharmacologic with pharmacologic treatments obtain better results.
- What methods exist that are not yet used in clinical practice, but could help in the assessment and management of pain?
We have explained in 5.3.2 part that several objective methods evaluate the physiological response to the pain analysing different parameters (heart rate, skin conductance, measurement of the diameter of the pupil). They could quantify the intensity of the pain and the response to the treatment. These parameters are objectives, non-invasive, relatively easy to use and could be put on at the patient´s bedside. However, they are unspecific (other non-pain stimulus like agitation, fear, stress produce the same response) and are expensive. For this reason, they are no used in clinical practice.

Reviewer 4 Report
Thank you for the review on Pain in Fetus neonates and children.
Few suggestions
1. Please give a little bit more explanation on developmental aspect of Pain. a little bit is described in fetal pain section but can be applicable to the neonatal pain too
2. Give references for the dosage and half life of the pharmacological agents. Or explain which drug formulary (Micromedex/neofax, lexicomp etc ) was used as there are some minor differences
Author Response
Reviewer 4
- Please give a little bit more explanation on developmental aspect of Pain. a little bit is described in fetal pain section but can be applicable to the neonatal pain too
According to the recommendations of the reviewer I have included an explanation of the development of the pain
- Give references for the dosage and half life of the pharmacological agents. Or explain which drug formulary (Micromedex/neofax, lexicomp etc ) was used as there are some minor differences
The dosage of pharmacological agents are based in several references. I have included the references in the text

Round 2
Reviewer 1 Report
The author responded appropriately to my suggestions in this revision. Mainly, I appreciate the information in Table 1 which adds more unique contributions to this field. I would suggest to accept the manuscript for publication.